# Integrated multimodality microscope for accurate and efficient target-guided cryo-lamellae preparation

Weixing Li [1,2,7], Jing Lu[1,7], Ke Xiao[1,7], Maoge Zhou[1,3,7], Yuanyuan Li[1,2], Xiang Zhang[1], Zhixun Li[4], Lusheng Gu [1,2,5], Xiaojun Xu[3], Qiang Guo [4,6]✉, Tao Xu [1,2,3,5]✉ & Wei Ji [1,2,5]✉

Cryo-electron tomography (cryo-ET) is a revolutionary technique for resolving the structure of subcellular organelles and macromolecular complexes in their cellular context. However, the application of the cryo-ET is hampered by the sample preparation step. Performing cryo-focused ion beam milling at an arbitrary position on the sample is inefficient, and the target of interest is not guaranteed to be preserved when thinning the cell from several micrometers to less than 300 nm thick. Here, we report a cryogenic correlated light, ion and electron microscopy (cryo-CLIEM) technique that is capable of preparing cryo-lamellae under the guidance of three-dimensional confocal imaging. Moreover, we demonstrate a workflow to preselect and preserve nanoscale target regions inside the finished cryo-lamellae. By successfully preparing cryo-lamellae that contain a single centriole or contact sites between subcellular organelles, we show that this approach is generally applicable, and shall help in innovating more applications of cryo-ET.

Observing cellular events in their native state is a direct and important way to ultimately understand how life works. Cryo-ET[1] is currently the principal technique for investigating the detailed structural biology of subcellular organelles and macromolecular complexes in situ[2–6]. However, the imaging depth of cryo-ET is restricted to a few hundred nanometers due to the limited penetration depth of electrons[7]. For high-resolution imaging, the sample thickness must be less than the inelastic mean free path of the electrons, which is approximately 300 nm in vitrified samples for the most widely used transmission electron microscopes (TEMs)[8]. Therefore, to investigate thicker samples, such as eukaryotic cells, thinning the specimen to lamellae in the thickness of roughly 200 nm or less is necessary.

Cryogenic-focused ion beam (cryo-FIB) milling is a recently developed method for preparing lamellae from vitrified biological samples for cryo-ET[9–12]. This technique avoids artifacts such as distortions, crevasses or compression when cryo-lamellae are prepared using traditional ultramicrotomy[13,14]. A big limitation of conventional FIB milling is the inability to determine the microfabrication region, and this is because both the FIB and scanning electron microscopy (SEM) images only deliver the surface view of the sample, without providing further information for recognition and localization of targets of interest (TOIs) buried underneath before milling. Therefore, this approach is only suitable for preparing lamellae of abundant cellular structures that are highly likely to be preserved when milled at an arbitrary position. For dispersed targets, fabricating thin lamellae while preserving the TOI is very challenging.

Correlative light and electron microscopy (CLEM)[15–18] has been developed to combine the specific labeling power of fluorescence

[1]Institute of Biophysics, Chinese Academy of Sciences, Beijing, China. [2]Bioland Laboratory (Guangzhou Regenerative Medicine and Health Guangdong Laboratory), Guangzhou, China. [3]Guangzhou Laboratory, Guangzhou International Bio Island, Guangzhou, China. [4]State Key Laboratory of Protein and Plant Gene Research, Peking-Tsinghua Center for Life Sciences, School of Life Sciences, Peking University, Beijing, China. [5]College of Life Science, University of Chinese Academy of Sciences, Beijing, China. [6]Changping Laboratory, Beijing, China. [7]These authors contributed equally: Weixing Li, Jing Lu, Ke Xiao, Maoge Zhou. ✉e-mail: guo.qiang@pku.edu.cn; xutao@ibp.ac.cn; jiwei@ibp.ac.cn

microscopy (FM) with the high spatial resolution of electron microscopy (EM), allowing the identification and localization of fluorescently labeled biological events on the TEM images[19–24] and SEM images[25]. Recently, FM has been combined with dual-beam FIB–SEM systems to locate the fluorescent TOI on the FIB image, which facilitates the FIB microfabrication of cryo-lamellae at specific sites[26–30]. In a conventional pipelined approach, in which FM and FIB–SEM are conducted successively in separated instruments[26], it is necessary to transfer samples among microscopes and intermediate cryo-workstations. This makes the workflow time consuming and introduces risks of sample devitrification and ice contamination. A recently reported integrated FM and FIB–SEM system has demonstrated the capability to perform FM-guided FIB milling in the same vacuum chamber[27]. This integrated approach simplifies the operation of the CLEM experiment, and enables real-time FM inspection of the sample without the need to transfer the sample between FM and FIB–SEM. To date, most integrated FM–FIB–SEM systems, including commercially available systems such as integrative fluorescence light microscopy (Thermo Fisher Scientific)[28] and METEOR (Delmic)[29], mainly use widefield microscopy for fluorescence imaging and use fiducial markers to register the FM and FIB images. However, the lack of the axial information in these integrated FM–FIB–SEM systems hinders the precise three-dimensional (3D) localization of both the TOIs and fiducials, making it challenging to perform accurate FIB milling of small subcellular structures. Moreover, fiducial-based image registration requires enormous effort and complicated algorithms for fiducial recognition and/or selection and coordinate transformation[26,31,32]. The concentration and the distribution of the fiducials also need to be carefully controlled to deliver repeatable and accurate correlation results, and these steps add more complexity to the experiment.

In this Article, we report a new cryogenic correlated light, ion and EM (cryo-CLIEM) system that incorporates a 3D multicolor confocal microscope into a dual-beam FIB–SEM system. In addition, we developed a dedicated workflow to prepare cryo-lamellae that contain specific TOIs under the guidance of light microscopy (LM). Our approach features several advantages over existing solutions, including the following: (1) integrated confocal microscopy with a high numerical aperture (NA) (up to 0.9) objective provides images of whole cells with abundant subcellular information in 3D, allowing the on-site precise localization, preselection and preservation of TOI inside of the finished cryo-lamellae; (2) CLIEM uses an FIB-etched benchmark instead of conventional fiducial markers to correlate the LM and FIB images, and this technique simplifies the sample preparation and delivers high reproducibility; (3) the 3D confocal image is projected to a two-dimensional (2D) image along the FIB milling angle, and the projected image, named 'LM via FIB', is directly paired with the FIB image and is used to guide the FIB milling. This new image registration principle is far more efficient compared to the conventional fiducial-based coordinate transformation; (4) the high-resolution LM image of the prepared cryo-lamellae could be used to navigate cryo-ET data collection. Moreover, by correlating the fluorescence signal with the tomographic reconstruction, CLIEM also provides insightful information for the identification and localization of specific structures in the cryo-ET data analysis; and (5) the flexible optical and mechanical design of the integrated confocal microscope can be extended to a wide range of commercial FIB–SEM systems as an add-on module, and we have also proved the compatibility of our confocal module with FIB–SEM instruments from Zeiss and Thermo Fisher Scientific by 3D modeling. Using all these advances, we demonstrated the applications of CLIEM by investigating lipid droplet (LD)–mitochondria interactions, mitochondria-endoplasmic reticulum contact (MERC) and the microtubule-organizing center of mammalian cells in CLIEM-prepared cryo-lamellae using cryo-ET, which are all challenging tasks to perform with conventional cellular tomography.

## Results

### Design and implementation of CLIEM

We integrated a custom-built confocal microscope into a commercial dual-beam FIB–SEM system (Tescan S8000G) (Fig. 1, Supplementary Figs. 1 and 2 and Supplementary Video 1). The confocal scanning arm was mounted on an additional panel that was attached to the front door of the FIB–SEM, without the need of modifying the mechanical constructions of the original door. The confocal detecting arm and the laser combiner were outsourced onto a separate optical table to reduce the weight added to the suspended system. These two units were connected to the confocal scanning arm via optical fibers, which did not hinder the opening of the chamber door. An optical window was embedded in a vacuum flange and was mounted in an existing port of the vacuum chamber to allow light to pass through. The objective was installed vertically in the vacuum chamber on a piezo-stage for fine focusing and z-scanning. This arrangement prevented the objective from interfering with other modules and reduced the risk of objective contamination from platinum (Pt) coating or FIB milling. The objective did not require retraction or calibration during operation, which simplified the experiment and increased the system stability. The confocal microscope was equipped with customized six excitation lasers and three parallel detection channels for multicolor imaging. In addition to fluorescence imaging, we also established simultaneous bright-field imaging that was based on the back-reflected light to capture the surface profile of the sample. With this setup, we were able to use off-the-shelf high NA objectives to image vitrified biological samples on EM grids with a high optical quality (Supplementary Fig. 3 and Supplementary Note 1). Confocal imaging also enabled the magnification to be flexibly adjusted and delivered a maximum field of view (FOV) of 145 × 145 μm (Supplementary Fig. 3e). To transfer cryogenic samples, we exploited a standard TEM multispecimen cryo-holder (Gatan 910) and developed a special mechanical interface to load/unload the sample onto/from the cryostage (Supplementary Fig. 4, Supplementary Note 2 and Supplementary Video 2). We modified the Gatan holder to accommodate three AutoGrid samples (Fig. 1d) and minimized the working distance of LM imaging to 1 mm by proper mechanical design (Fig. 1e).

In a CLIEM experiment, the sample was shifted among the following working positions: sample loading position, LM position and FIB–SEM position (Fig. 1a,c and Supplementary Video 1). In the sample loading and LM positions, the specimen lay horizontally, whereas during FIB milling, the stage was tilted to deliver a lower milling angle. Slots were carved on the sample holder to allow the FIB beam to pass through (Supplementary Fig. 4a,b), which enabled the milling on all the three AutoGrids. The LM and FIB–SEM positions were programmatically coupled so that the same TOI could be maintained within the FOV after the position was switched, allowing the fluorescence to be instantly checked during FIB milling. The FIB–SEM system was controlled by the original Tescan Essence software, and C++-based software was developed to control the stage movement and acquire LM images.

### Workflow of CLIEM

Instead of fiducial-based coordinate transformation, we established a more efficient method to register the LM and FIB images (Fig. 2a). To establish a registration benchmark, we etched a cross-shaped reference pattern (RP) beside the TOI (typically on the grid bar), which took roughly 1 min to perform. After acquiring confocal images of the interested region, we projected the reconstructed 3D image onto a 2D plane along the direction of FIB milling (Methods), which was termed 'LM via FIB'. The LM via FIB image could be easily correlated to the FIB image based on the position of the RP, thereby transforming the complicated registration between a 3D LM image and a 2D FIB image that were taken from different angles into a straightforward 2D correlation from the same view.

Based on these innovations, we developed a dedicated workflow to perform LM-guided FIB milling of vitrified samples using CLIEM. To demonstrate the work procedure, we froze roughly 10 nm sized

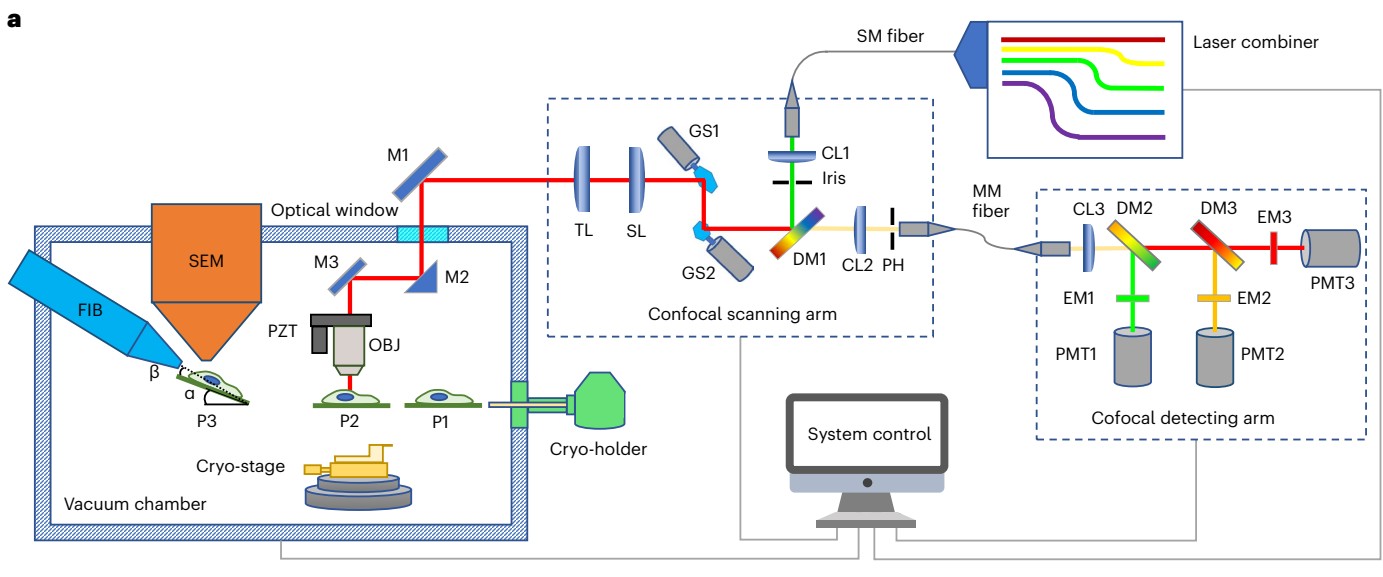

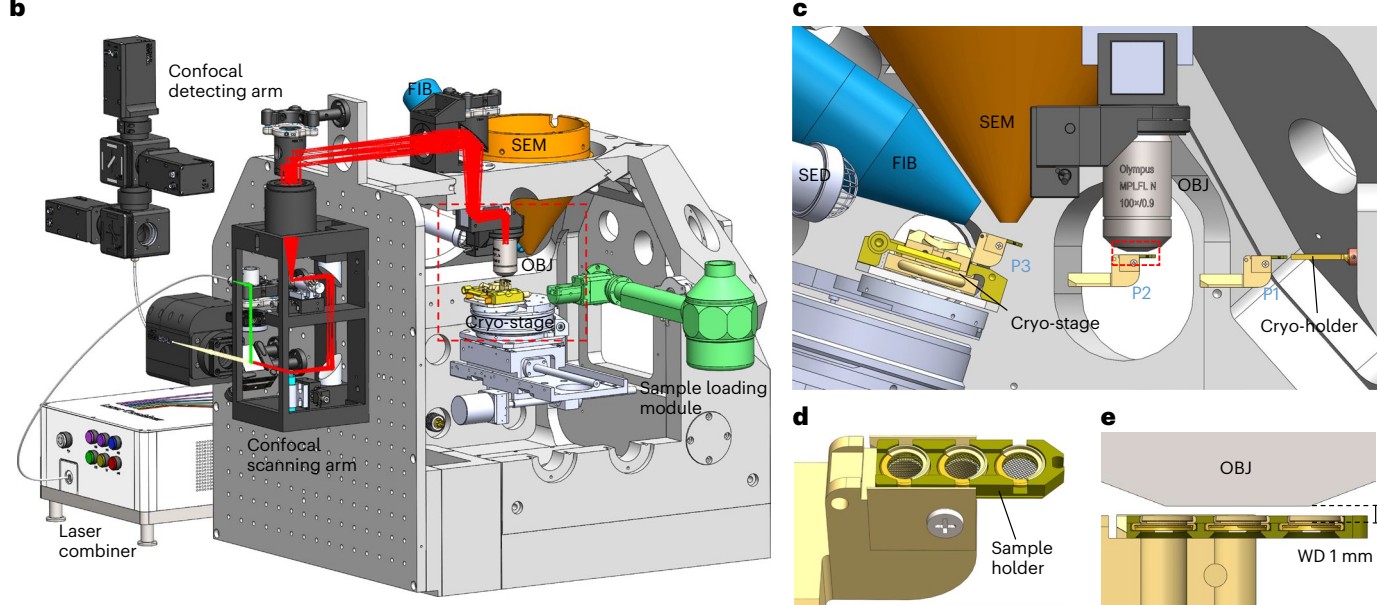

**Fig. 1 | Implementation of CLIEM. a**, Schematic diagram of the CLIEM system. α, Stage tilting angle during FIB milling; β, milling angle between the FIB and the sample plane; P1–P3, sample loading position (P1); LM imaging position (P2) and FIB–SEM position (P3). Abbreviations: objective (OBJ), piezo-stage (PZT), mirror (M), tube lens (TL), scanning lens (SL), galvo scanner (GS), collimating lens (CL), dichroic mirror (DM), pinhole (PH), single-mode (SM) fiber, multimode (MM) fiber and emission filter (EM). **b**, 3D rendering of the CLIEM system, showing the arrangement of the essential components and modules. **c**, The front view of the boxed area in **b**, illustrating the sample holder resting at three working positions (P1–P3). SED, secondary electron detector. **d**, 3D rendering of the sample holder containing three AutoGrid samples. **e**, The section view of the boxed area in **c**, displaying the 1 mm working distance (WD) of the confocal imaging module.

quantum dots (QDs) in ice on the EM grid to simulate the TOIs and tried to prepare cryo-lamellae while preserving single QDs. First, the sample was screened by SEM and confocal imaging, and an appropriate milling region containing the QDs was selected. Then, the sample was sputter-coated with Pt using a gas injection system (GIS) to form a protective layer on the surface, and the RP was etched beside the selected region. Afterward, 3D confocal images of this region were acquired. On the projected LM via FIB image, we measured the distance between a selected QD (as the TOI) and the center of the RP (for pixel size calibration, see Supplementary Note 1). Using the measured distance, we located the QD on the FIB image with respect to the same RP (Fig. 2b(i)). This method for target localization did not require the LM and FIB images to be superimposed, which simplified the correlation process. The localization precision in this step was determined by the

pixel sizes of the two imaging modalities, as well as the accuracy of the manual selection of the TOI and RP center positions. Typically, we used an FOV of 84 × 84 μm to cover both the TOI and the RP and acquired LM and FIB images with 1,536 × 1,536 pixels, yielding a pixel size of 54 nm in both imaging modalities. The error of manual position selection was typically 2–4 pixels, resulting in an overall localization precision of approximately 200 nm.

The localization precision only describes how precisely a TOI can be located on the FIB image. To accurately mill the TOI, mechanical sample drift must be considered. In our system, the sample drift was typically around 100 nm min$^{-1}$ along the milling direction, resulting in an approximately 50–600 nm drift during milling processes, which took approximately 0.5–6 min to perform. To mill the TOI with a high accuracy, we introduced an FM-guided milling process in two steps.

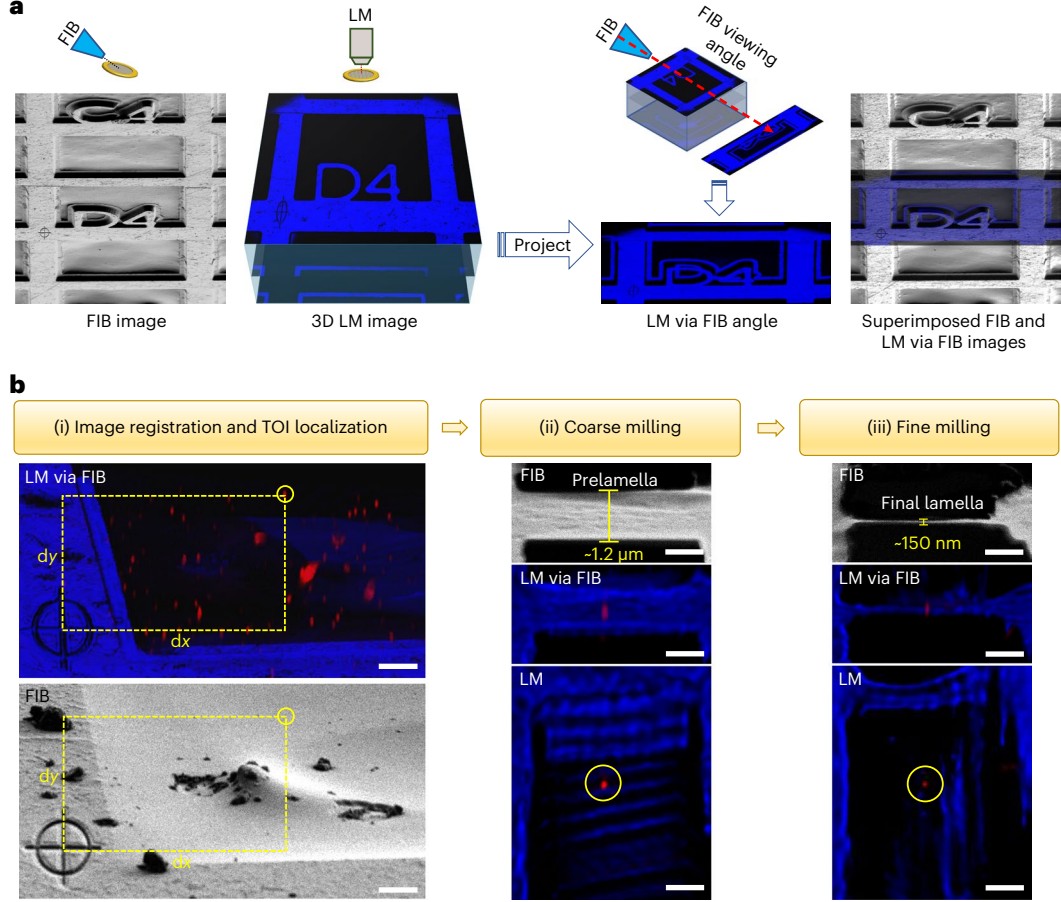

**Fig. 2 | Working principle and workflow of CLIEM. a**, Registration of FIB and LM images based on the FIB-etched RP and the LM image projection along the FIB angle. The superimposed FIB and LM via FIB images show their likeness, indicating that the projection method is valid. **b**, LM-guided FIB milling procedure. (i) Registration of the LM via FIB image (upper) and the corresponding FIB image (lower), showing the correlative localization of a selected QD as the TOI (circled) by measuring its distances to the same RP in each imaging modality. (ii),(iii), FIB and LM images of the coarsely milled prelamella (ii) and the finely milled final lamella (iii), respectively. The QD was preserved in the lamella during the whole milling process. Scale bars in **b**, 5 μm (i) and 1 μm (ii) and (iii).

In the first step (Fig. 2b(ii)), a prelamella of roughly 1.3 μm thick was fabricated at the determined position after image registration. This prelamella tolerated the moderate registration precision as well as the relatively large sample drift (approximately 600 nm) during the coarse milling process (that took approximately 6 min to finish), guaranteeing the presence of the QD in the prelamella. In the second step (Fig. 2b(iii)), the prelamella was imaged by the LM again and the position of the QD to the lamella boundaries was determined (Supplementary Fig. 5). This position guided us to precisely trim from the upper and lower sides of the prelamella, leaving a final lamella of roughly 150 nm thick containing the QD, which was confirmed by the LM inspection of the prepared lamella.

### In situ cryo-ET of subcellular organelle contact sites

Organelles physically interact with each other through membrane contact sites to coordinate the physiological activity of the cells[33]. Interorganelle membrane contacts are organized by tethers, which mainly consist of proteins to bridge the two organellar membranes[33]. Cryo-ET has been used to investigate the morphologies of the interfaces between organelles in situ[34]. By using CLIEM, we successfully prepared cryo-lamellae containing these contact sites with high efficiency, and we observed multiple tethering structures at high resolution between LDs and mitochondria, as well as in MERC.

LDs are highly dynamic organelles that store lipids in cells. Accumulating studies have shown that the interactions between LDs and mitochondria are essential in lipid metabolism and energy homeostasis[35]. EM has been a gold standard in resolving the LD–mitochondria contact that is in a typical range of less than 30 nm. Because LDs are relatively rare under normal physiological conditions, it is challenging to locate the contact sites with EM. We established a fluorescent reporter of LD–mitochondria contact by attaching the split fragments of superfolder green fluorescent protein (GFP) to the LD and the mitochondria outer membrane. The two fragments reconstituted into a complete GFP and became fluorescent when LDs and mitochondria came in close proximity, thus enabling the visualization of LD–mitochondria contact in LM[36] (Supplementary Fig. 6).

To preview the lamellae contents, we computationally generated 'virtual lamellae' from the 3D confocal image at various milling positions (Fig. 3a–d and Supplementary Video 3). We chose the optimal milling position in which the signal of the contact sites was clearly visible (Fig. 3c), and we prepared the lamella at this position (Fig. 3e). The LM image of the prepared lamella (Fig. 3f) was consistent with the LM image of the corresponding virtual lamella (Fig. 3c), proving the validity of this approach. In addition, the LM image of the prepared lamella helped us to identify the TOI on the TEM image (Fig. 3g) by superimposing these two images (Fig. 3h), which facilitated the cryo-ET data collection. After tomographic reconstruction, the LDs and mitochondria were identified and their contact sites were resolved (Fig. 3i,j and Supplementary Video 4). At most contact sites, we observed tethering structures that connected the LDs and the mitochondria (Fig. 3k and Extended Data Fig. 1). Furthermore, we correlated the LM image of the cryo-lamella with the tomogram and found that the fluorescence signal consistently

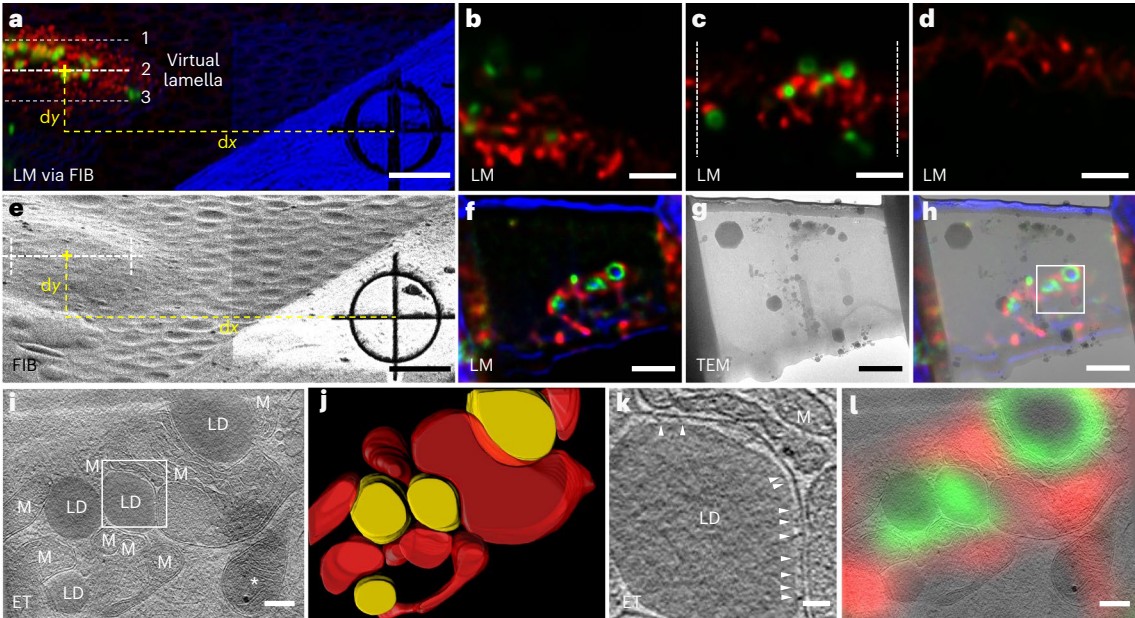

**Fig. 3 | LM-guided FIB milling and cryo-ET of LD–mitochondria contact sites in HepG2 cells. a**, LM via FIB of HepG2 cells grown on an EM grid, showing three milling positions of the virtual lamella as examples. The green channel shows LD–mitochondria contact sites that were genetically labeled with superfolder GFP. The red channel shows the mitochondria labeled with MitoTracker Deep Red. The blue channel shows the bright field. **b**–**d**, Computationally generated confocal images of the virtual lamellae at the three positions in **a**: 1 (**b**), 2 (**c**) and 3 (**d**). The zone between the dashed lines in position 2 was chosen for FIB milling. For the complete virtual lamellae images through the whole cell, see Supplementary Video 3. **e**, FIB image of the same region as in **a**, illustrating the determination of the milling position derived from the measured dx and dy in **a**. **f**, Confocal image of the prepared lamella, showing consistency with the virtual image in **c**. **g**, 300 kV TEM image of the prepared lamella. **h**, Superimposition of the LM image (**f**) and TEM image (**g**) of the prepared lamella, providing guidelines on choosing the region (boxed) for cryo-ET data collection. **i**, A tomographic slice of the HepG2 cell showing the LDs surrounded by mitochondria (M). An ice crystal is marked with an asterisk. **j**, 3D rendering of the LDs (yellow) and the mitochondrial outer membrane (red). **k**, Tomogram of the boxed region in **i**, showing the contact between an LD and a mitochondrion. Tethers at the contact site are indicated by arrows. **l**, Correlation between the LM image of the prepared lamella in **f** and the tomographic slice in **i**, showing the agreement between the fluorescence signal and the reconstructed structures. Five cryo-ET experiments were repeated independently with similar results. Scale bars are 5 μm in **a** and **e**; 2 μm in **b**–**d**, **f**–**h**; 200 nm in **i** and **l** and 50 nm in **k**.

overlapped with the expected structures (Fig. 3l). This result indicated that confocal images could also be used to assist in the identification and segmentation of specific structures in the cryo-ET data.

Using the same method as that used for the LD–mitochondria experiments, we investigated the MERC as well to demonstrate the general applicability of CLIEM. MERC is the best characterized inter-organelle membrane contact and functions as a central hub for a variety of biological processes, such as exchange of lipids and calcium, and mitochondria fission and fusion[33]. By establishing a fluorescent reporter of the MERC using split-GFP[36] (Supplementary Fig. 7), we visualized the MERC in HepG2 cells in LM and prepared lamellae containing MERC. Using cryo-ET, we resolved MERC and observed the tethering structures between the endoplasmic reticulum and the mitochondrial outer membrane as well (Extended Data Fig. 2).

**In situ cryo-ET of centrosomes in HeLa cells**
The centrosome is an microtubule-organizing center that is present in most animal cells. It plays a crucial role in cell division, polarity regulation, signaling and various other biological processes[37]. The centrosome consists of a pair of centrioles that are surrounded by pericentriolar material. Normally, each cell maintains only one or two centrosomes in different phases of the cell cycle. Therefore, it has been challenging to investigate the in situ structure of the centrosome by cryo-ET. To target the centrosome, we used genetically modified HeLa cells that expressed mCherry-labeled pericentrin[38], which is located in the pericentriolar material region of the centrosome. In LM, the centrosomes appeared as diffraction-limited spots (Fig. 4a), which could be easily located on the FIB image after registration (Fig. 4b). Using the two-step milling workflow, as described previously, we successfully prepared

cryo-lamellae that contained a single centrosome (Fig. 4c), and its position was conveniently located on the TEM image (Fig. 4d) using the fluorescence signal.

After the tomographic reconstruction was performed, we observed a centriole that was surrounded by abundant cellular structures in the native state and resolved the classic ninefold symmetrical arrangement of microtubule triplets (MTTs) (Fig. 4e). From the projected cross-section of the centriole (Fig. 4g), we observed a ring structure, which had a diameter of approximately 100 nm and 27 evenly distributed density nodes on the ring. These density nodes displayed rod-like shapes with a diameter of approximately 7 nm and a length of approximately 25 nm (Fig. 4h and Supplementary Video 5), which were similar to the densities identified in a previous study[39]; however, the functions of these shapes were not clear. Furthermore, using a subtomogram averaging[40] approach, we determined the polarity of each surrounding microtubule and confirmed that, as expected, most of the microtubules were grown out from the centriole[41] (Fig. 4f and Supplementary Video 5).

## Discussion

In summary, we have developed an integrated CLIEM system and a dedicated workflow to prepare cryo-lamellae of vitrified biological samples containing specific TOIs for in situ cryo-ET. Our approach features high accuracy and high efficiency and is easy to use. We demonstrated that CLIEM can not only determine the optimal milling position, such as organelle contact sites, but is also capable of targeting and preparing cryo-lamellae of rare biological structures or events, such as centrosomes. Furthermore, we showed that the LM image of the prepared lamella can assist in locating the TOI in TEM and provide

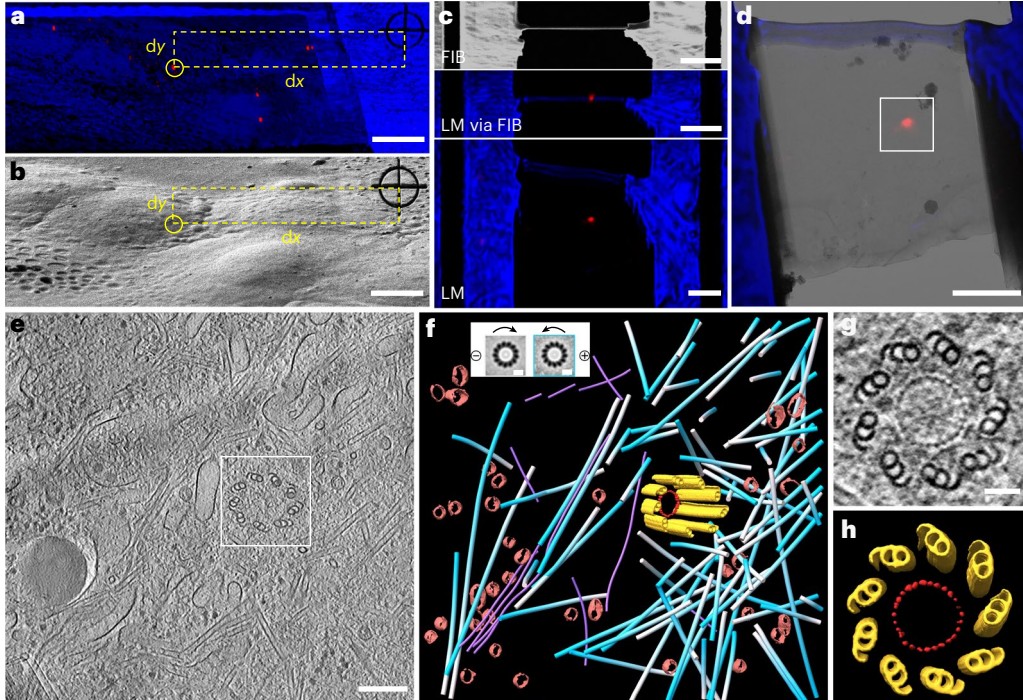

**Fig. 4 | LM-guided FIB milling and cryo-ET of centrosome in HeLa cells. a**, LM via FIB of HeLa cells grown on an EM grid, showing the distance measurement between a selected centrosome (circled) and the RP. The red channel shows a centrosome genetically labeled with mCherry. The blue channel shows the bright field. **b**, FIB micrograph of the same region as in **a**, illustrating the determination of the milling position according to the measured d$x$ and d$y$ in **a**. **c**, FIB and LM images of the prepared lamella with a thickness of roughly 200 nm, containing the targeted single centrosome. **d**, Superimposition of the LM and 300 kV TEM images of the prepared lamella. The boxed area was chosen for cryo-ET data collection. **e**, A tomographic slice showing a centriole surrounded by other cellular structures. **f**, 3D rendering of the centriole MTTs (yellow), ring structure in the centriole (red), surrounding microtubules (blue–white), intermediate filaments (violet) and transport vesicles (brown). Insert, the polarity determination of the microtubules, color-coded as white and blue for negative and positive orientation, respectively. **g**, A tomographic slices of the cross-section of the centriole in **e** in the top view, showing the symmetrical ninefold MTT arrangement and a ring structure with 27 evenly distributed density nodes. **h**, 3D rendering of the centriole in **g** based on STA. Three cryo-ET experiments were repeated independently with similar results. Scale bars are 10 μm in **a** and **b**; 3 μm in **c** and **d**; 200 nm in **e**; 10 nm in the **f** insert and 50 nm in **g**.

complementary information about the species and locations of specific biomolecules in the tomogram.

Our approach of integrating a confocal microscope into an FIB–SEM as an add-on module is generally applicable to a wide range of commercial FIB–SEM systems. Our design does not require that the objective has a particular orientation or installation position, which makes the integration flexible in the crowded vacuum chamber. The LM module can be further improved in several aspects to increase the performance and extend the functionalities. (1) The confocal detectors can be upgraded from photomultiplier tubes (PMTs) to avalanche photodiodes or hybrid detectors to achieve a higher detection sensitivity[42]. (2) Confocal imaging can be modified to Airyscan[43] or stimulated emission depletion[44] imaging to further improve the optical resolution. (3) Single molecule localization microscopy[45] can be added to CLIEM to provide versatile imaging capabilities, allowing specific molecules to be localized in a cryo-tomogram[46].

CLIEM introduces a new efficient way of performing LM-guided FIB milling due to its multicolor 3D imaging capability and integrated design: (1) as the benchmark for image registration, FIB-etched RP can completely replace the fiducial markers that are randomly added to the sample before vitrification[47]. (2) The registration principle based on 'LM via FIB' projection and direct distance measurement circumvents the conventional time-consuming procedure and complicated algorithms using coordinate transformation[26,31,32]. This new registration process can be accomplished using the ready-to-use plugin (Volume Viewer) in open-source software (Fiji[48]) with minimal effort, and it can be adapted by other existing CLEM workflows to improve their performances. (3) CLIEM allows for an instant FM inspection of the prepared cryo-lamellae before

sending them to the TEM, which increases the efficiency of cryo-ET experiments. (4) Besides high efficiency, CLIEM also delivers high success rate of targeted-FIB milling. We prepared in total 42 cryo-lamellae of biological samples in this work, and 40 of them contained the desired feature (Extended Data Fig. 3), which resulted in an overall success rate of 95% (Supplementary Table 2). For the MERC and centrosome experiments, 100% prepared lamellae contained the desired targets.

We also demonstrated that the high-resolution LM image of the final lamella was of great use for cryo-ET experiments. By correlating the LM and TEM images of the final lamella, it became straightforward and convenient to locate the desired targets in the crowded and low-contrast TEM image. This is particularly useful when investigating targets that are not clearly visible in a low magnification TEM image, such as protein complexes or phase separations. In the future, the relatively slow manual correlation between LM and TEM images can be automated by developing corresponding image processing software for more efficient and convenient operation.

CLIEM can be applied to a wide range of biological systems. Our method brings new insights into the organization of interorganelle membrane contacts. In the future, CLIEM could be used to perform the structural analysis of tethering proteins in situ at high resolution, hence promoting the understanding of interorganelle membrane contacts under physiological and pathological conditions. For point-like TOIs, the position of which can be determined by finding the maxima of its point spread function with high precision[49], as demonstrated in the QD and centrosome experiments in this work. Therefore, CLIEM could also benefit the in situ investigation of cellular ultrastructures such as viruses, small vesicles and protein complexes.

CLIEM adopts a standard TEM cryo-holder to transfer cryogenic samples manually. In the future, developing a dedicated autoloader for CLIEM that is compatible with cryo-TEM could benefit this method and improve the user experience and working efficiency. And the introduction of super-resolution FM could further improve the performance of CLIEM. In conclusion, CLIEM has the potential to serve as an all-in-one solution for cryo-ET sample preparation and promote the developments of in situ structural biology[50].

## Online content

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

## Methods

### Integrating confocal microscopy into an FIB–SEM system

Mechanical design of CLIEM was performed using SolidWorks software (2019, Dassault Systems), and optical design of the integrated confocal microscope was performed using Zemax OpticStudio software (16.5 SP5, Zemax) (Supplementary Fig. 2a,b). Off-axis parabolic mirrors were used as relay optics between two galvo scanners to prevent pupil drifting without introducing chromatic aberrations[51,52]. Since the design parameters of commercial objectives were difficult to acquire, a paraxial lens with a corresponding focal lens was used to simulate the objective.

Design details are shown in Supplementary Fig. 2c. The integrated confocal microscope was composed of a custom-developed laser combiner, a confocal scanning arm, a confocal detecting arm and an objective module inside the vacuum chamber. The laser light from a multiwavelength laser combiner (405, 488, 532, 561, 638 nm) was directed through a single-mode fiber into the scanning arm that was attached to the FIB–SEM chamber door. The laser light was collimated by a collimator (CL1) and reflected by a dichroic mirror (DM1), scanned horizontally and vertically by two galvo scanners (GS1, GS2) and then relayed by a scan lens and a tube lens. The scanned illuminating beam passed through an optical window and reached the objective mounted within the vacuum chamber. The commercially available dry objective was mounted on a vacuum compatible piezo-stage, enabling 3D confocal imaging in the FIB–SEM chamber. To accommodate the limited traveling range of the sample stage, the optical path was folded by M1, M2 and M3 so that the sample could be moved to the area under the objective. The light emitted from the sample followed the same pathway as the exciting light, passed through DM1 and was focused by a collecting lens (CL2) before being passed through a pinhole, the size of which could be adjusted to optimize the resolution and light efficiency. After that, the reflected light and fluorescent light were coupled into a multimode fiber and entered the detecting arm. The detecting arm comprised three PMTs (PMT1, PMT2 and PMT2), two dichroic mirrors (DM2, DM3) and three emission filters (EM1, EM2 and EM3) dependent on the imaging channels. A bright-field imaging channel that used the reflected light from the sample and two fluorescent channels were used to precisely register the LM image and FIB image. To accommodate various imaging modes, the dichroic mirrors and emission filters could be switched as desired.

### Sample preparation

**QD.** QDs (QD625, NajingTech) were diluted at 1,000× in distilled water. Then 14 µl of QD solution was added onto 200 mesh copper EM grids (BZ10022F1, Beijing Zhongjingkeyi Technology Co., Ltd) for vitrification.

**LD–mitochondria.** HepG2 cells were cultured at 37 °C with 5% CO$_2$ in Dulbecco's modified Eagle's medium (DMEM, Gibco, catalog no. c11995500BT) supplemented with 10% fetal bovine serum (FBS, Gibco, catalog no. 16000-044) and 1% penicillin-streptomycin (HyClone, catalog no. 30010). A GFP1-10 sequence was inserted into the N terminal of human PLIN2. The resulting GFP1-10-*Plin2* was then inserted into a pCDH-CMV-MCS-EF1-Puro vector. The N terminal of human TOM20 (1-33aa) was used as a mitochondria targeting sequence. Mito-GFP11 was inserted into a pCDH-CMV-MCS-EF1-Puro vector. Lentivirus was packaged as previously described[53]. Wildtype HepG2 cells were infected by lentiviruses to stably express mito-GFP11 and GFP1-10-*Plin2*. The LD–mitochondria contact reporter cells were selected and maintained in DMEM supplemented with 2 µg ml$^{-1}$ puromycin. Then, the cells were grown on lacey 200 mesh gold EM grids (T10012Au, Beijing XXBR Technology Co., Ltd) overnight to promote adherence and spreading. Mitochondria were stained with 1 µM MitoTracker Deep Red (M22426, Thermo Fisher Scientific) at 37 °C for 30 min. For the validation of the LD–mitochondria contact reporter, LDs were stained with HCS LipidTox red (H34476, Thermo Fisher Scientific) at a dilution of 1:1,000 at 37 °C for 30 min.

**MERC.** HepG2 cells were cultured and seeded onto EM grids as described above. To generate the MERC reporter, human SEC61B sequence was used as endoplasmic reticulum membrane targeting sequence. GFP1-10-*Sec61b* sequence was then inserted into a pCDH-CMV-MCS-EF1-Puro vector. Wildtype HepG2 cells were infected by lentiviruses to stably express mito-GFP11 and GFP1-10-*Sec61b*. The MERC reporter cells were selected and maintained in DMEM supplemented with 2 µg ml$^{-1}$ puromycin. Mitochondria were stained with 1 µM MitoTracker Deep Red (M22426, Thermo Fisher Scientific) at 37 °C for 30 min. For the validation of the MERC, the cells were transiently transfected with ER-DsRed for endoplasmic reticulum labeling[54].

**Centrosome.** HeLa cells that expressed mCherry in pericentrin were cultured at 37 °C with 5% CO$_2$ in DMEM supplemented with 10% FBS and 1% penicillin-streptomycin. The cells were seeded onto the lacey 200 mesh gold EM grids as described above.

The EM grids used in all experiments were plasma-cleaned in Gatan Solarus (950, Gatan) for 90 s before use. The grids used for cell experiments were subsequently sterilized under ultraviolet light for 30 min.

### Sample vitrification

All the samples were plunge-frozen into liquid ethane using a Leica EM GP2 automatic plunge-freezer. Cryogen temperature was set to −183 °C, and the chamber temperature was set to 37 °C with 85% humidity. Grids were blotted from the backside using Whatman Type 1 paper for 0.5 and 5 s for the QD experiments and the cell experiments, respectively. The vitrified grids were clipped into AutoGrids (Thermo Fisher Scientific) and mounted onto a multispecimen cryo-holder (Gatan 910) for further sample loading.

### LM imaging

LM imaging was performed using the integrated confocal microscope and custom-developed control hardware and software. The synchronization of GS1, GS2, piezo-stage and image acquisition was controlled by an I/O device (PXIe-6341, National Instruments), and the multi-channel signals from PMTs were acquired using a DAQ device (PXIe-6396, National Instruments) in parallel. Coarse movement of the sample was accomplished by moving the original motorized stage of the FIB–SEM system, while fine focusing and *z*-scanning were accomplished by moving the objective using the piezo-stage. A commercial objective (MPLFLN ×100/NA 0.9, Olympus) was used for all LM imaging. Depending on the signal intensity and the purpose of the image, different FOV, scanning speeds, stepping sizes and pinhole sizes were chosen during image acquisition. In all bright-field imaging experiments, a 561 nm laser was used for illumination and PMT2 with a neutral density filter (NE01B-A, Thorlabs) was used for detection. For fluorescence imaging, different excitation wavelengths and filter combinations in the detecting arm were applied as follows.

In the QD experiments, a 488 nm laser was used for excitation, and PMT3 with EM3 (ZET405/488/561m, Chroma) was used for fluorescence detection. In the LD–mitochondria and MERC experiments, for imaging the contact sites that were labeled with superfolder GFP, a 488 nm laser was used for excitation and PMT1 with EM1 (ZET405/488m, Chroma) was used for fluorescence detection; for imaging the mitochondria that were labeled with MitoTracker Deep Red, a 638 nm laser was used for excitation and PMT3 with EM3 (ZET405/488/561/640m, Chroma) was used for fluorescence detection. In the centrosome experiments, the 561 nm laser that was used for bright-field imaging was also used for fluorescence excitation and PMT3 with EM3 (ZET405/488/561/640m, Chroma) was used for fluorescence detection.

### LM image processing

**Image projection and distance measurement.** The LM images were projected onto a 2D plane along the FIB milling direction using open-source software Fiji[48] (v.1.53f51, Volume Viewer plugin). A Tricubic

sharp interpolation algorithm in the Volume Viewer plugin was applied on the projected image to generate cubic voxels. On the produced 'LM via FIB' images, the distance between the center of RP and the center of TOI was measured in pixels and converted into distance through pixel calibration (Supplementary Note 1).

**Virtual lamellae generation.** Virtual lamellae images were generated using Fiji's Volume Viewer plugin as well. First, the 3D LM images were rotated perpendicular to the FIB angle, and Slice Mode was selected for the display window. Then, by adjusting the distance of the slice, a virtual lamella at the corresponding position was generated.

**Image deconvolution.** For displaying purposes of the LM images used in this paper, LM images were deconvoluted using the Huygens software (v.20.04, Scientific Volume Imaging). Templates were selected according to the imaging parameters and signal-to-noise ratio to improve the resolution and contrast without introducing artifacts.

### Pt coating and FIB milling
The EM grids were coated with a 2–3 μm thick layer of Pt using GIS, which formed a protective layer on the surface to avoid lamellae damage during FIB milling. The coating was applied for 1–2 min at a GIS temperature of 30 °C, and the sample was placed 2 mm below the FIB–SEM coincidence point during coating.

FIB milling was conducted using various beam currents ranging from 10 to 500 pA. In particular, 500 pA was used to etch RPs and to perform coarse milling, 150 and 50 pA were used for fine milling and 10 pA was used for final polishing. The stage was tilted by 17° during milling, resulting in a milling angle of 18° between the FIB and the sample. Micro-expansion joints[55] were applied for cell lamellae to release the internal tension and decrease lamellae bending or crack.

### Cryo-ET data collection
All cryo-ET data were acquired on a Titan Krios cryo-TEM (Thermo Fisher Scientific) equipped with a BioQuantum energy filter and a K2 Summit Direct electron detector (Gatan). The TEM was operated at 300 kV in low-dose mode in all experiments, and tilt series data were collected unidirectionally using SerialEM v.3.8 software[56]. Detailed parameters of data acquisition are listed in Supplementary Table 1.

### Registration of cryo-LM and cryo-TEM images
The cryo-LM and cryo-TEM images were first scaled to the same magnification in accordance with the pixel size of each imaging modality. The scaled images were then manually registered according to the edge information of the lamella in the bright-field channel of the LM image and the TEM image. Registration only involved rotation and translation without introducing nonrigid deformations.

### Cryo-ET reconstruction
All tilt series frames were motion-corrected using Motioncor2 (ref. 57) software. The produced tilt series was aligned with the patch-tracking method and back projected to obtain the tomogram in the IMOD v.4.11.0 software package[58]. For segmentation, tomograms were rescaled with a binning factor of four. A deconvolution filter was applied on the tomogram to improve the contrast[59].

### Segmentation and visualization
**LD–mitochondria.** The lipid membrane and mitochondrial outer membrane were manually segmented and polished using Imaris software (v.9.8.0, Oxford Instruments). The segmentations were rendered and displayed using Imaris as well.

**Centrosome.** For the centriole, MTTs were manually traced with B-tubules centered using IMOD. Subtomograms were extracted along each triplet with a spacing of 4 nm using the RELION[60] helix toolbox,

resulting in 339 subtomograms in total. Subtomogram averaging was performed with RELION v.2.1 (ref. 61).

A total of 111 microtubules were automatically traced with Amira[62] followed by manual polishing. To prevent the effect of missing wedges, we applied helical symmetry during subtomogram averaging. As most microtubules contain 13 protofilaments[63], we assumed the rise and twist to be 9.23 Å and 27.69°, respectively. Then, 50,580 subtomograms were then cropped with Dynamo v.1.1.532 software[64] from bin2 tomograms. The microtubule polarity was determined using a previously published method[65]. In brief, with Dynamo, two iterations of shift search were applied without any angular search, which was followed by three iterations of the first two Euler angles (phi and theta) search and two iterations of the third Euler angles (psi) search. The shift search range along the microtubule was set to 1 pixel, and the in-plane rotation was limited to 27°. Then, we projected the averaging results along the filament to generate 2D images to determine the polarity.

Membrane segmentation was established with a tensor voting-based method[66], followed by manual polishing in Amira. Only transport vesicles were retained for visualization. Segmentations were rendered and displayed with ChimeraX v.1.3 software[67].

### Reporting summary
Further information on research design is available in the Nature Portfolio Reporting Summary linked to this article.

## Data availability
The tomographic reconstructions in this work have been deposited in the Electron Microscopy Database with the accession codes EMD-33496 (LD–mitochondria) and EMD-33495 (centrosome). Other source data that support the findings of this study are available from the corresponding author upon request.

## Code availability
The software for stage control and image acquisition, which is hardware-dependent, is available from the corresponding author upon request.

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

## Acknowledgements

We are grateful to F. Sun from Institute of Biophysics (IBP), Chinese Academy of Sciences (CAS) for providing kind suggestions on our work and generous supports in the system development. We appreciate the help from J. Chen and F. Wang from Peking University on providing the HeLa cells expressing mCherry in centrosome. We thank X. Huang, J. Zhang, B. Zhu and S. Li from the Center for Biological Imaging (CBI), CAS on cryo-ET data collection; P. Wang from IBP and T. Niu from CBI for assisting in the cryo-ET data reconstruction; Y. Teng and Y. Feng from CBI for their help on cryo-ET data segmentation and visualization and L. Liu from CBI for helping with plunge freezing. We also thank P. Liu and X. Zhang from IBP for providing kind suggestions on the interpretation of the LD–mitochondria data; K. He from Institute of Genetics and Developmental Biology, CAS for providing SUM159 cells stably expressing EGFP-Rab5c; S. Gu from University of Electronic Science and Technology of China for the discussion on image processing of the prelamella. We also recognize the assistance of Q. Wang from IBP in photographing the system. This work was supported by the National Key Research and Development Program of China (grant no. 2021YFA1301500 to W.J.); the National Natural Science Foundation of China (grant nos. 32027901 to T.X. and 92254306 to W.J.); the National Science Foundation for Distinguished Young Scholars of China (grant no. T2225020 to W.J.); the National Key Research and Development Program of China (grant no. 2022YFC3400600 to W.J.); the Strategic Priority Research Program of the CAS (grant no. XDB37040104 to W.J.); the Youth Innovation Promotion Association of CAS (grant nos. 2013066 to W.J., 2017135 to L.G. and 2021088 to Y.L.); Beijing Nova Program from Beijing Municipal Science & Technology Commission (grant no. Z202003 to L.G.); the National Natural Science Foundation of China (grant nos. 31700743 to L.G., 32170704 to L.G., 32100536 to M.Z., 62105356 to J.L. and 32001075 to Y.L.); the Scientific Instrument Development Project of the CAS (grant no. GJJSTD20210001 to T.X.). This work is also an output of the National Project of Multi-mode, Multi-scale Biomedical Imaging.

## Author contributions

W.J., T.X., Q.G., W.L. and J.L. designed the experiments. W.J., W.L. and J.L. developed the CLIEM system. X.Z. wrote the control software. X.X. designed and constructed the laser combiner. M.Z. and Y.L. conducted cell culture and labeling. K.X. performed sample vitrification, cryo-lamellae preparation and cryo-ET data collection. W.L., J.L. and K.X. perform LM image processing. W.L. and Z.L. performed the cryo-ET data analysis and image processing. T.X., W.J., Q.G., W.L., J.L., K.X. and L.G. interpreted the results. W.J., T.X., Q.G., W.L. and J.L. wrote the manuscript, which was modified by all other authors.

## Competing interests

A CN patent (ZL 201920857325.X) has been issued describing the sample loading module used in this work; W.L., W.J. and T.X. are the coinventors. A CN patent describing the integration of a confocal microscope into an FIB–SEM system has been filed (202110469220.9); T.X., J.L., W.J., W.L., K.X. and X.Z. are the coinventors. A CN patent describing the CLIEM workflow has been filed (202211206312.9); T.X., W.J., W.L., J.L. and K.X. are the coinventors. The remaining authors declare no competing interests.

## Additional information

**Extended data** is available for this paper at https://doi.org/10.1038/s41592-022-01749-z.

**Correspondence and requests for materials** should be addressed to Qiang Guo, Tao Xu or Wei Ji.

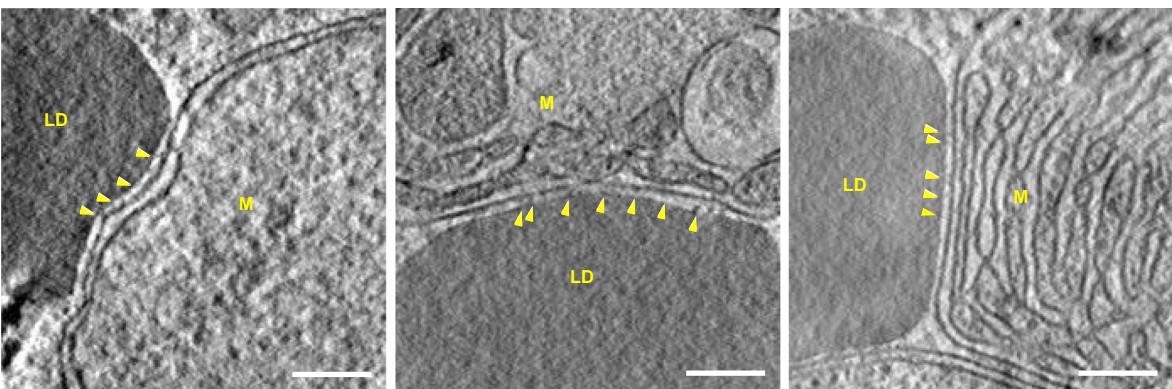

**Extended Data Fig. 1 | Tethering structures at the LD–mitochondria contact sites.** Tomograms of LD–mitochondria contact sites from three different HepG2 cells, showing the tethering structures (arrows) between the LD and mitochondria (M). Scale bar is 100 nm. Five cryo-ET experiments were repeated independently with similar results.

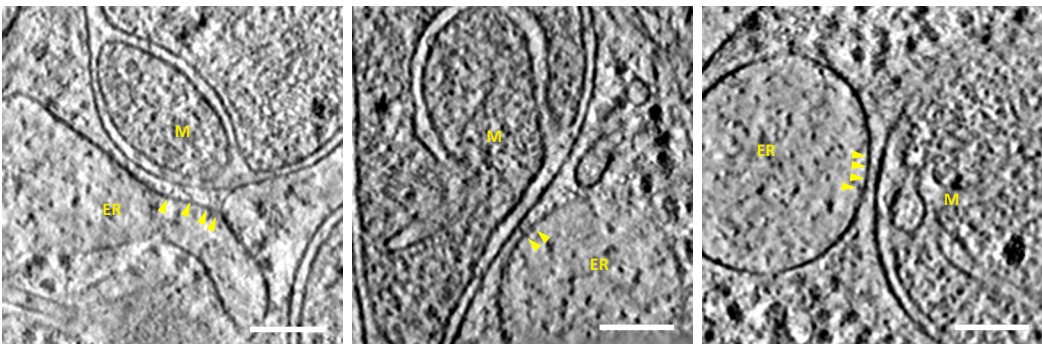

**Extended Data Fig. 2 | Tethering structures of MERC.** Tomograms of MERC from different HepG2 cells, showing the tethering structures (arrows) between the ER and mitochondria (M). Scale bar is 100 nm. Three cryo-ET experiments were repeated independently with similar results.

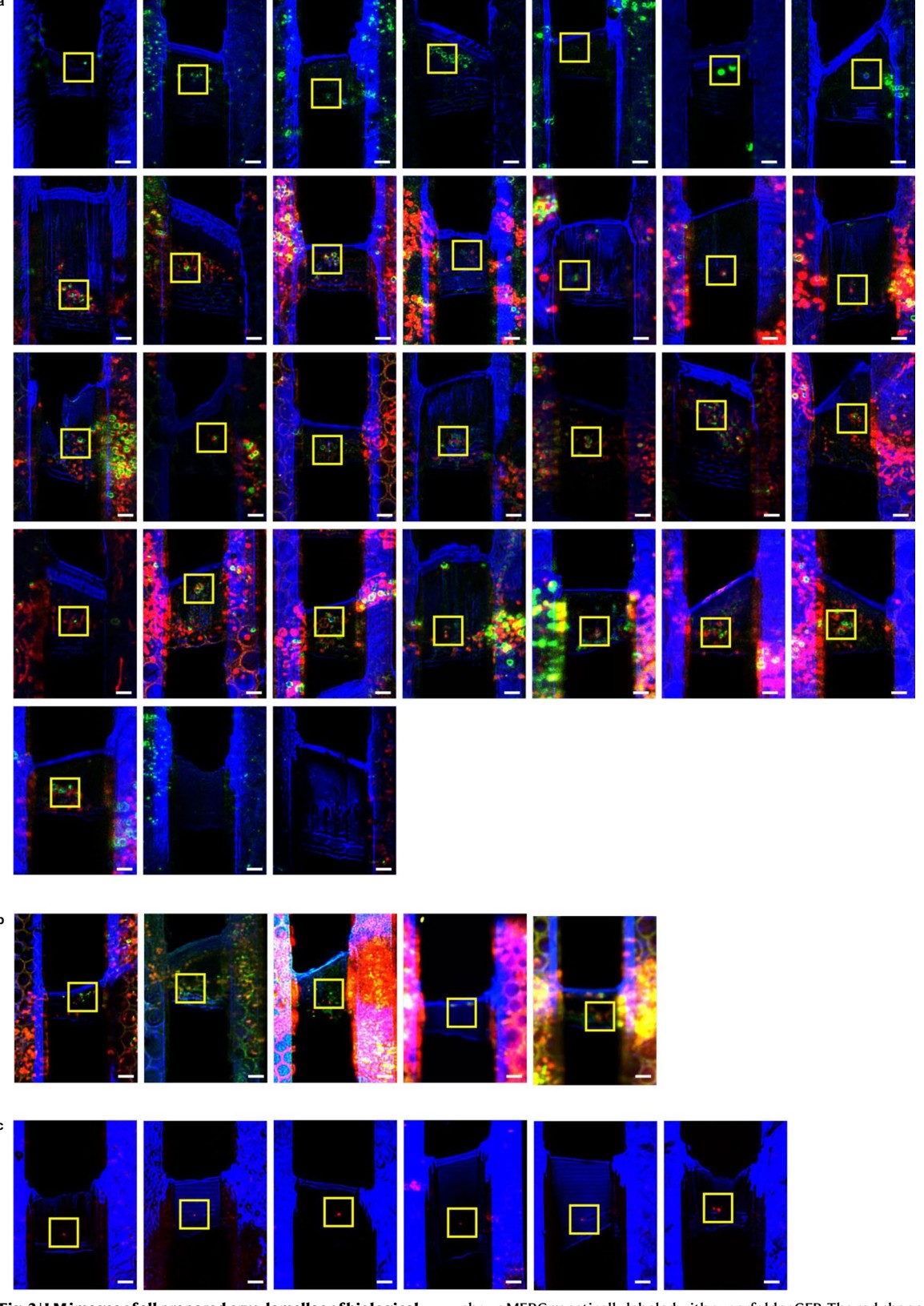

**Extended Data Fig. 3 | LM images of all prepared cryo-lamellae of biological samples. a**, 31 lamellae of LD–mitochondria contact sites in HepG2 cells with 2 failed milling. The green channel shows LD–mitochondria contact sites genetically labeled with superfolder GFP. The red channel shows mitochondria labeled with MitoTracker Deep Red. The blue channel shows the bright field. **b**, 5 lamellae of MERC in HepG2 cells with 0 failed milling. The green channel shows MERC genetically labeled with superfolder GFP. The red channel shows mitochondria labeled with MitoTracker Deep Red. The blue channel shows the bright field. **c**, 6 lamellae of centrosome in Hela cells with 0 failed milling. The red channel shows centrosome genetically labeled with mCherry. The blue channel shows the bright field. Yellow boxes indicate the desired targets. Images without yellow box indicate failed milling. Scale bar is 2 μm.

# nature research

# Reporting Summary

Nature Research wishes to improve the reproducibility of the work that we publish. This form provides structure for consistency and transparency in reporting. For further information on Nature Research policies, see our Editorial Policies and the Editorial Policy Checklist.

## Statistics

For all statistical analyses, confirm that the following items are present in the figure legend, table legend, main text, or Methods section.

| n/a | Confirmed | |
|---|---|---|
| ☐ | ☒ | The exact sample size (*n*) for each experimental group/condition, given as a discrete number and unit of measurement |
| ☒ | ☐ | A statement on whether measurements were taken from distinct samples or whether the same sample was measured repeatedly |
| ☒ | ☐ | The statistical test(s) used AND whether they are one- or two-sided<br>*Only common tests should be described solely by name; describe more complex techniques in the Methods section.* |
| ☒ | ☐ | A description of all covariates tested |
| ☒ | ☐ | A description of any assumptions or corrections, such as tests of normality and adjustment for multiple comparisons |
| ☒ | ☐ | A full description of the statistical parameters including central tendency (e.g. means) or other basic estimates (e.g. regression coefficient) AND variation (e.g. standard deviation) or associated estimates of uncertainty (e.g. confidence intervals) |
| ☒ | ☐ | For null hypothesis testing, the test statistic (e.g. *F*, *t*, *r*) with confidence intervals, effect sizes, degrees of freedom and *P* value noted<br>*Give P values as exact values whenever suitable.* |
| ☒ | ☐ | For Bayesian analysis, information on the choice of priors and Markov chain Monte Carlo settings |
| ☒ | ☐ | For hierarchical and complex designs, identification of the appropriate level for tests and full reporting of outcomes |
| ☒ | ☐ | Estimates of effect sizes (e.g. Cohen's *d*, Pearson's *r*), indicating how they were calculated |

*Our web collection on statistics for biologists contains articles on many of the points above.*

## Software and code

Policy information about availability of computer code

| Data collection | TESCAN Essence 1.1.4.0 for system control and FIB/SEM image acquisition, C++ 11 with custom code for system control and LM image acquisition, SerialEM 3.8 for cryo-ET data collection. |
|---|---|
| Data analysis | SolidWorks 2019 for mechanical design, Zemax OpticStudio 16.5 SP5 for optical design, Fiji 1.53f51 for image analysis, measurePSF code in MATLAB R2017b for PSF analysis, Huygens 20.04 for LM image deconvolution, IMOD 4.11.0 for tomographic reconstruction, RELION v2.1 and Dynamo 1.1.532 for subtomogram averaging, Imaris 9.8.0 for segmentation and visualization, ChimeraX 1.3 for 3D rendering. |

For manuscripts utilizing custom algorithms or software that are central to the research but not yet described in published literature, software must be made available to editors and reviewers. We strongly encourage code deposition in a community repository (e.g. GitHub). See the Nature Research guidelines for submitting code & software for further information.

## Data

Policy information about availability of data

All manuscripts must include a data availability statement. This statement should provide the following information, where applicable:
- Accession codes, unique identifiers, or web links for publicly available datasets
- A list of figures that have associated raw data
- A description of any restrictions on data availability

The tomographic reconstructions in this work have been deposited in the Electron Microscopy Database (EMDB) with the accession codes EMD-33496 (LD-mitochondria), EMD-33495 (centrosome). Other source data that support the findings of this study are available from the corresponding author upon request.

# Field-specific reporting

Please select the one below that is the best fit for your research. If you are not sure, read the appropriate sections before making your selection.

☒ Life sciences ☐ Behavioural & social sciences ☐ Ecological, evolutionary & environmental sciences

For a reference copy of the document with all sections, see nature.com/documents/nr-reporting-summary-flat.pdf

# Life sciences study design

All studies must disclose on these points even when the disclosure is negative.

| | |
|---|---|
| Sample size | The number of lamellae prepared for LD-mitochondria, MERC and centrosome experiments is n=31, n=5, n=6, respectively. The number of cryo-ET tilt series collected for LD-mitochondria, MERC and centrosome experiments is n=5, n=3, n=3, respectively. We determined these sample sizes to be sufficient because the targets of interests could be resolved with desired resolution using cryo-ET. |
| Data exclusions | No data was excluded from the study. |
| Replication | For LM-guided FIB-milling, the number of final lamellae that contained the desired structures after FIB-milling is n=29, n=5, n=6 for LD-mitochondria, MERC and centrosome experiments, respectively, resulting in an overall 95% replication rate. For cryo-ET experiments, all collected cryo-ET tilt series, in total n=5, n=3, n=3 for LD-mitochondria, MERC and centrosome experiments, respectively, delivered desired target structures. |
| Randomization | Cells were randomly selected for lamellae preparation. Lamellae that contained minimal ice contamination and breakage were selected for tilt series data collection. |
| Blinding | Blinding is considered not necessary for the method, because the purpose of this work was to demonstrate a new technique, and this technique does not depend on the statistical variation of the properties of the samples. For the cell experiments, blinding was not possible because the experimental conditions were evident from the image data. |

# Reporting for specific materials, systems and methods

We require information from authors about some types of materials, experimental systems and methods used in many studies. Here, indicate whether each material, system or method listed is relevant to your study. If you are not sure if a list item applies to your research, read the appropriate section before selecting a response.

## Materials & experimental systems

| n/a | Involved in the study |
|---|---|
| ☒ | ☐ Antibodies |
| ☐ | ☒ Eukaryotic cell lines |
| ☒ | ☐ Palaeontology and archaeology |
| ☒ | ☐ Animals and other organisms |
| ☒ | ☐ Human research participants |
| ☒ | ☐ Clinical data |
| ☒ | ☐ Dual use research of concern |

## Methods

| n/a | Involved in the study |
|---|---|
| ☒ | ☐ ChIP-seq |
| ☒ | ☐ Flow cytometry |
| ☒ | ☐ MRI-based neuroimaging |

# Eukaryotic cell lines

Policy information about cell lines

| | |
|---|---|
| Cell line source(s) | HepG2 cell line (1101HUM-PUMC000035, National Infrastructure of Cell Line Resource, China). Hela cell line (ATCC, CCL-2) was provided from Prof. Jianguo Chen's lab at Peking University. |
| Authentication | The cell lines were not further authenticated in our lab. |
| Mycoplasma contamination | All cell lines used in this study tested negative for mycoplasma contamination. |
| Commonly misidentified lines (See ICLAC register) | No commonly misidentified lines were used in the study. |

