## [Peer Review File · Nature Methods]

Peer Review Information

Manuscript Title: Integrated multimodality microscope for high-accuracy and high-efficiency target-guided cryo-lamellae preparation

Corresponding author name(s): Qiang Guo, Tao Xu, Wei Ji

Editorial Notes: n/a

Reviewer Comments & Decisions:

Decision Letter, initial version:

Dear Wei,

Please let me begin by apologizing again for our delay during the peer review process. We have decided to move forward with our decision based on two reviewer reports.

Your Article, "Integrated multimodality microscope for high-accuracy and high-efficiency target-guided cryo-lamellae preparation", has now been seen by two reviewers. As you will see from their comments below, although the reviewers find your work of considerable potential interest, they have raised a number of concerns. We are interested in the possibility of publishing your paper in Nature Methods, but would like to consider your response to these concerns before we reach a final decision on publication.

We therefore invite you to revise your manuscript to address these concerns. We ask that you address the technical concerns calling for improved quantitative characterizations of performance. We are not concerned about novelty in this case, but we do ask that you make sure that your work is placed in the proper context and that related works are appropriately cited and discussed. We do not require that you add a demonstration on smaller cellular components, but if such data are readily available, we do agree that they would strengthen the paper.

[Redacted] This URL links to your confidential home page and associated information about manuscripts you may have submitted, or that you are reviewing for us. If you wish to forward this email to co-authors, please delete the link to your homepage.

We hope to receive your revised paper within 2-3 months. If you cannot send it within this time, please let us know. In this event, we will still be happy to reconsider your paper at a later date so long as nothing similar has been accepted for publication at Nature Methods or published elsewhere.

OPEN SCIENCE REQUIREMENTS

REPORTING SUMMARY AND EDITORIAL POLICY CHECKLISTS

Please note that these forms are dynamic ‘smart pdfs’ and must therefore be downloaded and completed in Adobe Reader. We will then flatten them for ease of use by the reviewers. If you would like to reference the guidance text as you complete the template, please access these flattened versions at <http://www.nature.com/authors/policies/availability.html>.

DATA AVAILABILITY

All novel DNA and RNA sequencing data, protein sequences, genetic polymorphisms, linked genotype and phenotype data, gene expression data, macromolecular structures, and proteomics data must be deposited in a publicly accessible database, and accession codes and associated hyperlinks must be provided in the “Data Availability” section.

Please include a “Data availability” subsection in the Online Methods. This section should inform readers about the availability of the data used to support the conclusions of your study, including accession codes to public repositories, references to source data that may be published alongside the paper, unique identifiers such as URLs to data repository entries, or data set DOIs, and any other statement about data availability. At a minimum, you should include the following statement: “The data that

support the findings of this study are available from the corresponding author upon request”, describing which data is available upon request and mentioning any restrictions on availability. If DOIs are provided, please include these in the Reference list (authors, title, publisher (repository name), identifier, year). For more guidance on how to write this section please see: <http://www.nature.com/authors/policies/data/data-availability-statements-data-citations.pdf>

CODE AVAILABILITY

Please include a “Code Availability” subsection in the Online Methods which details how your custom code is made available. Only in rare cases (where code is not central to the main conclusions of the paper) is the statement “available upon request” allowed (and reasons should be specified).

For more information on our code sharing policy and requirements, please see: <https://www.nature.com/nature-research/editorial-policies/reporting-standards#availability-of-computer-code>

MATERIALS AVAILABILITY

ORCID

Nature Methods is committed to improving transparency in authorship. As part of our efforts in this direction, we are now requesting that all authors identified as ‘corresponding author’ on published papers create and link their Open Researcher and Contributor Identifier (ORCID) with their account on the Manuscript Tracking System (MTS), prior to acceptance. This applies to primary research papers

only. ORCID helps the scientific community achieve unambiguous attribution of all scholarly contributions. You can create and link your ORCID from the home page of the MTS by clicking on 'Modify my Springer Nature account'. For more information please visit www.springernature.com/orcid.

Sincerely,
Rita

Rita Strack, Ph.D.
Senior Editor
Nature Methods

Reviewers' Comments:

Reviewer #1:

Remarks to the Author:

This manuscript describes a novel FIB/SEM microscope with an integrated confocal light path, which allows the user to target specific features within vitrified cells prior to lamella preparation. The workflow is a two-step process where the sample is first imaged as a confocal z-stack, and then virtual lamellae are projected and overlaid on the FIB image for guided milling. The authors then use the instrument to target three different objects as proof of its capabilities: 1) lipid droplet-mitochondrial contacts, 2) mitochondria-endoplasmic reticulum contacts and 3) a microtubule organizing center. It is a beautiful display of the potential for this new CLIEM instrument.

That being said, I have some concerns about its suitability for Nature Methods:

1) While the integration of a confocal beam path is unique, this basic approach is not fundamentally novel. A similar approach was used by Arnold et al, 2016 in the Biophysical Journal, minus the virtual lamella generation.

Arnold, Jan, et al. "Site-specific cryo-focused ion beam sample preparation guided by 3D correlative microscopy." *Biophysical journal* 110.4 (2016): 860-869.

2) Without comparing the performance of this current method to previous approaches, it is impossible to judge the efficiency of the workflow. How many of each target were attempted and how many were a success?

In my opinion, this paper was fun and easy to read. While the general CLIEM approach may not be entirely novel, the integrated workflow most certainly is. The novelty of investigating virtual lamellae prior to milling is elegant. If concern 2 was addressed, and the approach indeed increases the efficiency of targeted milling significantly, I would suggest this manuscript for publication in *Nature Methods*.

Reviewer #2:

None

Reviewer #3:

Remarks to the Author:

This paper presents an advanced system by adding a confocal microscope into a conventional SEM/FIB system in order to define the target of interest during cryo-FIB milling more accurately, especially with its unique 3D information achieved from 3D confocal imaging. The breakthrough in this new method is the "LM via FIB" which is quite original and impressive for the application of cryo CLEM in the future, but there is no strong evidence to show its unique advantage that the 3D confocal information became the key to achieve the result either for LD-mitochondria or centrosome. This paper needs much more specific experiments to prove that the 3D confocal information could offer high resolution fluorescence signal to enable distinguishing particular target out of crowded cellular contents, to thus support the application ability of their method. Other than the major concern above there are a few aspects the authors may want to consider in the future:

(1) The paper needs to present a quantitative improvement of success ratio since it is the key or hot topic for cryo FIB milling.

(2) What's the correlation accuracy between FM and cryoEM? The methods need to have this question addressed in order to highly correlate the target for cryoET to achieve high resolution.

(3) It would be more attractive if the paper could focus on smaller cellular components less than 100nm. By that in my opinion the paper is not sufficient for this journal.

Author Rebuttal to Initial comments

Reviewers' Comments:

Reviewer #1:

Remarks to the Author:

This manuscript describes a novel FIB/SEM microscope with an integrated confocal light path, which allows the user to target specific features within vitrified cells prior to lamella preparation. The workflow is a two-step process where the sample is first imaged as a confocal z-stack, and then virtual lamellae are projected and overlaid on the FIB image for guided milling. The authors then use the instrument to target three different objects as proof of its capabilities: 1) lipid droplet-mitochondrial contacts, 2) mitochondria- endoplasmic reticulum contacts and 3) a microtubule organizing center. It is a beautiful display of the potential for this new CLIEM instrument.

Response: We thank the reviewer for the positive comments.

That being said, I have some concerns about its suitability for Nature Methods:

- 1) While the integration of a confocal beam path is unique, this basic approach is not fundamentally novel. A similar approach was used by Arnold et al, 2016 in the Biophysical Journal, minus the virtual lamella generation. Arnold, Jan, et al. "Site-specific cryo-focused ion beam sample preparation guided by 3D correlative microscopy." Biophysical journal 110.4 (2016): 860-869.

Response: The reviewer is correct that the approach of using confocal imaging to guide FIB milling in two separate instruments has already been established by Arnold et al in 2016. We highly recognized this work and cited it in our manuscript (Line 66 and 68). And we also revised the main text in Line 78 to make the description clearer.

To our minds, our integrated solution brought the conventional approach a great step forward. We did not just simply integrate a confocal into a dual-beam SEM, we also made great use of the 3D imaging ability by introducing a conceptually new working routine that involved fiducial-free FIB-benchmarking, "virtual lamella" screening and two-step milling. By adopting our approach, we were able to achieve greater efficiency and more operational convenience. In this regard, we believe that our integrated approach is sufficiently significant to be introduced to the community.

- 2) Without comparing the performance of this current method to previous approaches, it is impossible to judge the efficiency of the workflow. How many of each target were attempted and how many were a success?

Response: We thank the reviewer for suggesting a comparison of the performance between our integrated method and conventional separated approaches. Based on all the cell experiments conducted in this study, we analyzed the success rate of our method and found that 95% of the targeted FIB-milling were successful (final lamella containing the desired target). We added Supplementary Table 2 to show the detailed data and statistics, and the corresponding LM images of the prepared lamellae were shown in Response Fig.

1. Our total success rate of 95% (100% for centrosome and MERC) is significantly higher than that of 60% in Arnold's approach (Arnold et al., Biophysical journal 110, 860-869 (2016)), which was more of an estimation based on the error calculation of the fiducial-based correlation accuracy without experimental approval. Notably, this success rate of 60% was concluded when aiming at a final lamella thickness of 300 nm. When preparing thinner lamellae, as in our case typically less than 200 nm, the success rate in Arnold's approach would be even lower.

In my opinion, this paper was fun and easy to read. While the general CLIEM approach may not be entirely novel, the integrated workflow most certainly is. The novelty of investigating virtual lamellae prior to milling is elegant. If concern 2 was addressed, and the approach indeed increases the efficiency of targeted milling significantly, I would suggest this manuscript for publication in Nature Methods.

Response: We thank the reviewer for the positive comments. We hope the reviewer could agree that we have addressed the concern 2 as above.

Response Figure 1. LM images of all prepared cryo-lamellae of biological samples in this work. a, 31 lamellae of LD-mitochondria contact sites in HepG2 cells with 2 failed milling. Green channel: LD-mitochondria contact sites genetically labeled with Superfolder GFP. Red channel: mitochondria labeled with MitoTracker Deep Red. Blue channel: BF. b, 5 lamellae of MERC in HepG2 cells with 0 failed milling. Green channel: MERC genetically labeled with Superfolder GFP. Red channel: mitochondria labeled with MitoTracker Deep Red. Blue channel: BF. c, 6 lamellae of centrosome in HeLa cells with 0 failed milling. Red channel: centrosome genetically labeled with mCherry. Blue channel: BF. Yellow boxes indicate the desired targets. Images without yellow box indicate failed milling. Scale bar, 2 μm .

Reviewer #2:
None

Reviewer #3:

Remarks to the Author:

This paper presents an advanced system by adding a confocal microscope into a conventional SEM/FIB system in order to define the target of interest during cryo-FIB milling more accurately, especially with its unique 3D information achieved from 3D confocal imaging. The breakthrough in this new method is the “LM via FIB” which is quite original and impressive for the application of cryo CLEM in the future, but there is no strong evidence to show its unique advantage that the 3D confocal information became the key to achieve the result either for LD-mitochondria or centrosome. This paper needs much more specific experiments to prove that the 3D confocal information could offer high resolution fluorescence signal to enable distinguishing particular target out of crowded cellular contents, to thus support the application ability of their method.

Response: We thank the reviewer for the positive comments. As the reviewer mentioned, one breakthrough of our method is the “LM via FIB” approach to locate the target accurately. This approach is only possible when using the 3D information of the sample, which was obtained in our case by 3D confocal imaging. Therefore, we believe that 3D imaging is necessary and is the key to achieving the 3D localization and accurate FIB-milling of specific subcellular structures in our work, elucidated as follows. For the centrosome experiments, 3D imaging allows for convenient and accurate localization of the point-like target in the “LM via FIB” image. To accomplish this with 2D imaging, one would have to either arrange the objective parallel to the FIB milling angle, which is usually impossible because of mechanical interference in the crowded vacuum chamber; or with the LM focal plane coincident with the FIB milling focal point. In the latter case, one way to conduct fluorescence-guided FIB milling is to observe the fluorescence signal of the target continuously during milling, and the milling should be stopped when the signal decays due to partial destruction of the target. Compared to these approaches based on 2D imaging, we believe that our approach is more advantageous and elegant, because it does not require specific orientation when installing the objective, nor does it introduce destruction to the target. The non-destructive milling is particularly meaningful when preparing lamella of small targets such as single viruses or protein complexes, which do not have much excess material that can be eliminated to observe signal decay. Additionally, to demonstrate the application of our method on smaller point-like targets, we did endosome experiments to prove that the 3D confocal could be used to guide the lamellae fabrication for targets with a size less than 100 nm (see Response to Concern 3). For the LD-mitochondria experiments, the contact sites were rather dense, and it was very difficult to distinguish particular contact sites and determine the best milling position upon the 2D “LM via FIB” image. Besides, the fluorescence decay method that works for point-like targets using 2D imaging does not work for these crowded targets. Taking advantage of the 3D information, we were able to “slice” the “LM via FIB”

image at arbitrary positions, and inspect the content on the “virtual lamella” to find particular targets in crowded cellular contents. As shown in Fig. 3, the fluorescence image of the final lamella (Fig. 3f) is consistent with the “virtual lamella” (Fig. 3c), which means that we could get the best lamella with

desired information using 3D imaging. Moreover, we did MERC experiments to prove that the 3D confocal imaging could offer high-resolution fluorescence signal in more crowded cellular contents (Supplementary Figure 8). Similar to LD-mitochondria experiments, we were able to select a particular MERC slice out of crowded context for FIB milling by using “virtual lamella” inspection in 3D. Furthermore, our method is advantageous over existing methods in several other aspects: (i) our method provided a high success rate of 95% for site-specific FIB milling, which was much higher than other approaches (see Response to concern 1); (ii) our method reduced the risk of sample damaging or contamination during the sample transfer in the pipelined methods; (iii) our 3D correlation approach avoided continuous irradiation by the excitation light as in approaches based on 2D imaging, thus avoiding potential photo-bleaching and sample warmup.

Other than the major concern above there are a few aspects the authors may want to consider in the future:

(1) The paper needs to present a quantitative improvement of success ratio since it is the key or hot topic for cryo FIB milling.

Response: We thank the reviewer for the suggestion. Based on all the cell experiments conducted in this study, we analyzed the success rate of our method and found that 95% (100% for centrosome and MERC) of the targeted FIB-milling were successful (final lamella containing the desired target). We added Supplementary Table 2 to show the detailed data and statistics, and the corresponding LM images of the prepared lamellae were shown in Response Fig. 1. Our success rate of 95% is significantly higher than that of 60% in Arnold’s approach (Arnold et al., Biophysical journal 110, 860-869 (2016)), which was more of an estimation based on the error calculation of the fiducial-based correlation accuracy without experimental approval. Notably, this success rate of 60% was concluded when aiming at a final lamella thickness of 300 nm. When preparing thinner lamellae, as in our case typically less than 200 nm, the success rate in Arnold’s approach would be even lower.

(2) What’s the correlation accuracy between FM and cryoEM? The methods need to have this question addressed in order to highly correlate the target for cryoET to achieve high resolution.

Response: As a fiducial-free method, the registration of the cryo-LM and cryo-TEM images in our CLIEM method was based on the edge information of the prepared lamella in both imaging modalities. In order to address the reviewer’s concern, we fabricated a pattern with two edges on the EM grids covered by supporting film to simulate the cryo-lamella, and we added fluorescent beads as benchmarks to evaluate the correlation accuracy. We correlated the cryo-LM and cryo-TEM images using our registration method (added in line 441-445), quantified the correlation accuracy by co-localizing the fiducial markers in both

image modalities, and concluded a correlation accuracy of less than 50 nm (mean \pm standard deviation = 28.94 ± 17.61 nm).

Method: Fluorescent beads with 200 nm diameter (T7280, ThermoFisher Scientific) were diluted and spread on EM grids (T10022F, Beijing XXBR Technology Co., Ltd.), which were then plunge-frozen and transferred into our CLIEM system. 9 lamellae containing in total 54 beads were prepared using FIB milling, and LM images of the lamellae were taken in our system (Response Fig. 2a). Then the sample was transferred into a Titan Krios cryo-TEM (ThermoFisher Scientific), and cryo-TEM images of the lamellae were taken (Response Fig. 2b). After that, cryo-LM and cryo-TEM images were manually registered according to the edge information in the BF channel of the LM image and the TEM image (Response Fig. 2c). In the aligned image, the centroids of the fiducial markers in the fluorescence channel of the LM image and the cryo-TEM image were determined by normal and flat top Gaussian fitting, respectively (Response Fig. 2d). We evaluated the correlation accuracy by calculating the deviation error between the centroid coordinates in cryo-LM and cryo-TEM images for each bead.

Results: We calculated the deviation error of the centroid positions of total 54 beads in 9 lamellae. The scatter plot of co-localization errors in x and y (Response Fig. 2e) showed that more than 85% of the beads were co-localized with an accuracy better than 50 nm. The co-localization errors in absolute distance were determined for each lamella (Response Fig. 2f), and the mean co-localization error for the total 54 beads was calculated as 28.94 ± 17.61 nm, which suggested an overall correlation accuracy of less than 50 nm as well. We believe that this accuracy is sufficient for fluorescence-guided cryo-ET data collection and the correlation of the LM and TEM images.

Response Figure 2. Evaluation of correlation accuracy between cryo-LM and cryo-TEM. a, Maximum projection of 3D cryo-LM image along the optical axis. b, Cryo-TEM image of the same region as in a. c, Correlated cryo-LM (a) and cryo-TEM (b) image. d, Centroid coordinates of the beads derived from the correlated image in c. e, Scatter plot of centroid co-localization errors in x and y for the total 54 beads in 9 lamellae. f, Mean and standard deviation of the distance between beads in cryo-LM and cryo-TEM images for each lamella. Scale bar, 5 μm .

(3) It would be more attractive if the paper could focus on smaller cellular components less than 100nm. By that in my opinion the paper is not sufficient for this journal.
 Response: Early endosome (EE) is a major sorting compartment for vesicles budding from plasma membrane via endocytosis. It has been demonstrated that EEs are usually $\sim 100\text{--}500$ nm in diameter (Klumperman J, et al. Cold Spring Harb Perspect Biol. May 22;6(10): a016857. (2014)). Rab5 is an EE marker and we labeled EEs with EGFP-Rab5c using CRISPR/Cas9-mediated genome editing (Kangmin He et al. Nature 552, 410–414 (2017)). Using our CLIEM system, we were able to locate single EE in 3D and prepare cryo-lamella containing the desired single EE at the cell periphery (Response Fig. 3a). Individual

endosomes shown progressively flows from small EEs at the cell periphery to large endosomes at the center (Roberto Villaseñor et al. *Current Opinion in Cell Biology* 39:53–60(2016)). After tomographic

reconstruction, we clearly resolved the targeted Rab5-positive EE with a diameter of ~ 80 nm (Response Fig. 3b), which might be the nascent very early EEs generated by recruiting Rab5 to uncoated endocytic vesicles (Kangmin He et al. *Nature* 552, 410–414 (2017)). These results suggested that our method is applicable to investigate small cellular structures less than 100 nm.

Response Figure 3. Preliminary result of EE imaging using CLIEM system. a, Correlated cryo-LM and cryo-TEM image of a cryo-lamella containing a single EE. b, A tomographic slice of the region in a with the targeted single EE (white arrow). Scale bars, 1 μm in a; 100 nm in b.

Decision Letter, first revision:

Dear Wei,

Thank you for submitting your revised manuscript "Integrated multimodality microscope for high-accuracy and high-efficiency target-guided cryo-lamellae preparation" (NMETH-A49304C). It has now been seen by the original referees and their comments are below. The reviewers find that the paper has improved in revision, and therefore we'll be happy in principle to publish it in *Nature Methods*, pending minor revisions to satisfy the referees' final requests and to comply with our editorial and formatting guidelines.

Please note, we do not ask for any additional experiments at this point, but we do ask you to add the rebuttal data requested by ref 1 into the main text and that you discuss as a limitation that manual registration can be a slow step in your process.

TRANSPARENT PEER REVIEW

Nature Methods offers a transparent peer review option for new original research manuscripts submitted from 17th February 2021. We encourage increased transparency in peer review by publishing the reviewer comments, author rebuttal letters and editorial decision letters if the authors agree. Such peer review material is made available as a supplementary peer review file. Please state in the cover letter 'I wish to participate in transparent peer review' if you want to opt in, or 'I do not wish to participate in transparent peer review' if you don't. Failure to state your preference will result in delays in accepting your manuscript for publication.

Thank you again for your interest in Nature Methods Please do not hesitate to contact me if you have any questions.

Sincerely,
Rita

Rita Strack, Ph.D.
Senior Editor
Nature Methods

ORCID

Reviewer #1 (Remarks to the Author):

The authors have satisfied my concerns and I think this paper should be published. In addition to the added supplemental table 2, I would like to see "response figure 1" included in the supplemental data. It is a helpful visual depiction of the quality and quantity of their lamellae.

Reviewer #3 (Remarks to the Author):

Dear authors,

Thanks for the author to have carefully looked into all my concerns and made detailed answers correspondingly.

For my concern (1), the author added new quantitative number for successful ration of all FIB milling experiment, together to answering the concern from Reviewer #2. I am very convinced by the number.

For my concern (2), the author provided new experiment and data for the correlation between FM and cryoEM, but it is at low magnification which utilizes the cutting edge and beads. As I know it is not enough to offer enough accuracy for localization of region of interest at high magnification at cryoEM, especially at the view magnification for cryoET data collection. So I will still hold on my point.

For my concern (3), the new data shows indeed a vesicle with diameter of 80nm, but under more extensively non-common situation with random shape structure in crowded cellular content it may still be challenging. The manual registration between FM and cryoET is also a limitation and slow down factor for the application.

So I am afraid that I would rather not change my mind for the publication of this paper.

Author Rebuttal, first revision:

Reviewers' Comments:

Reviewer #1:

Remarks to the Author:

The authors have satisfied my concerns and I think this paper should be published. In addition to the added supplemental table 2, I would like to see "response figure 1" included in the supplemental data. It is a helpful visual depiction of the quality and quantity of their lamellae.

Response: We thank the reviewer for the suggestion. "Response Figure 1" was presented as Extended Data Figure 3.

Reviewer #3:

Remarks to the Author:

Dear authors,

Thanks for the author to have carefully looked into all my concerns and made detailed answers correspondingly.

For my concern (1), the author added new quantitative number for successful ration of all FIB milling experiment, together to answering the concern from Reviewer #2. I am very convinced by the number.

Response: We thank the reviewer for the positive comments.

For my concern (2), the author provided new experiment and data for the correlation between FM and cryoEM, but it is at low magnification which utilizes the cutting edge and beads. As I know it is not enough to offer enough accuracy for localization of region of interest at high magnification at cryoEM, especially at the view magnification for cryoET data collection. So I will still hold on my point.

Response: The correlation between FM and cryoEM at low magnification only serves to find the target on the crowded and low-contrast cryoEM image upon the fluorescence signal. We think the accuracy of ~50 nm that we have shown in the last response is sufficient for this task. After switching to high magnification, the structure of target became visible and recognizable on the cryoEM image. Therefore, we do not need to locate the region of interest according to the fluorescence at high magnification any more.

For my concern (3), the new data shows indeed a vesicle with diameter of 80nm, but under more extensively non-common situation with random shape structure in crowded cellular content it may still be challenging. The manual registration between FM and cryoET is also a limitation and slow down factor for the application.

Response: The reviewer is right that it may be challenging to identify the target under some extreme situation when the target is surrounded by similar cellular structure. But we think the correlation accuracy is enough for most events. And the specific fluorescence labeling also can help to highlight and distinguish the target in a crowded cellular content, as we have shown in the LD-mitochondria and MERC experiments. The contact sites in these two examples had random shape. Because we labelled the contact sites with a different color with BiFC method (Green) than the surrounding content (Red), the distinct fluorescence signal enabled exact localization of the target in the 3D confocal image. The manual registration between FM and cryoET took approximately 10 min to accomplish, thus should not become a major limitation of time in the whole CLIEM+cryoET experiment. The relatively slow

manual correlation between LM and TEM images can be automated by developing corresponding image processing software for more efficient and convenient operation in the future (added in line 288-290).

Final Decision Letter:

Dear Wei,

I am pleased to inform you that your Article, "Integrated multimodality microscope for high-accuracy and high-efficiency target-guided cryo-lamellae preparation", has now been accepted for publication in Nature Methods. Your paper is tentatively scheduled for publication in our Feb or March print issue, and will be published online prior to that. The received and accepted dates will be May 24, 2022 and Dec 6, 2022. This note is intended to let you know what to expect from us over the next month or so, and to let you know where to address any further questions.

Once your paper is typeset, you will receive an email with a link to choose the appropriate publishing options for your paper and our Author Services team will be in touch regarding any additional information that may be required.

Please note that *Nature Methods* is a Transformative Journal (TJ). Authors may publish their research with us through the traditional subscription access route or make their paper immediately open access through payment of an article-processing charge (APC). Authors will not be required to make a final decision about access to their article until it has been accepted. [Find out more about Transformative Journals](https://www.springernature.com/gp/open-research/transformative-journals)

Authors may need to take specific actions to achieve [compliance](https://www.springernature.com/gp/open-research/funding/policy-compliance-faqs) with funder and institutional open access mandates. If your research is supported by a funder that requires immediate open access (e.g. according to [Plan S principles](https://www.springernature.com/gp/open-research/plan-s-compliance)) then you should select the gold OA route, and we will direct you to the compliant route where possible. For authors selecting the subscription publication route, the journal's standard licensing terms will need to be accepted, including [journal-](https://www.springernature.com/gp/open-research/policies/journal-)

policies">self-archiving policies. Those licensing terms will supersede any other terms that the author or any third party may assert apply to any version of the manuscript.

Your paper will now be copyedited to ensure that it conforms to Nature Methods style. Once proofs are generated, they will be sent to you electronically and you will be asked to send a corrected version within 24 hours. It is extremely important that you let us know now whether you will be difficult to contact over the next month. If this is the case, we ask that you send us the contact information (email, phone and fax) of someone who will be able to check the proofs and deal with any last-minute problems.

If, when you receive your proof, you cannot meet the deadline, please inform us at rjsproduction@springernature.com immediately.

Once your manuscript is typeset and you have completed the appropriate grant of rights, you will receive a link to your electronic proof via email with a request to make any corrections within 48 hours. If, when you receive your proof, you cannot meet this deadline, please inform us at rjsproduction@springernature.com immediately.

Once your paper has been scheduled for online publication, the Nature press office will be in touch to confirm the details.

Once your paper has been scheduled for online publication, the Nature press office will be in touch to confirm the details.

Content is published online weekly on Mondays and Thursdays, and the embargo is set at 16:00 London time (GMT)/11:00 am US Eastern time (EST) on the day of publication. If you need to know the exact publication date or when the news embargo will be lifted, please contact our press office after you have submitted your proof corrections. Now is the time to inform your Public Relations or Press Office about your paper, as they might be interested in promoting its publication. This will allow them time to

prepare an accurate and satisfactory press release. Include your manuscript tracking number NMETH-A49304D and the name of the journal, which they will need when they contact our office.

About one week before your paper is published online, we shall be distributing a press release to news organizations worldwide, which may include details of your work. We are happy for your institution or funding agency to prepare its own press release, but it must mention the embargo date and Nature Methods. Our Press Office will contact you closer to the time of publication, but if you or your Press Office have any inquiries in the meantime, please contact press@nature.com.

Nature Portfolio journals [encourage authors to share their step-by-step experimental protocols](https://www.nature.com/nature-research/editorial-policies/reporting-standards#protocols) on a protocol sharing platform of their choice. Nature Portfolio 's Protocol Exchange is a free-to-use and open resource for protocols; protocols deposited in Protocol Exchange are citable and can be linked from the published article. More details can found at www.nature.com/protocolexchange/about.

Best regards,

Rita